# OpenReview forum: "$\epsilon$-VAE: Denoising as Visual Decoding"
_ICLR.cc/2025/Conference — Submitted to ICLR 2025_

### Official Review · Reviewer_VC75 · 2024-10-23

**Soundness:** 2
**Presentation:** 2
**Contribution:** 2
**Rating:** 3
**Confidence:** 4

**Summary:**

This paper focuses on improving the VAE and demonstrates its effectiveness using a generation task. The authors optimize the decoder part, and the diffusion model iteratively denoises the data to recover the original. They compare two metrics, rFID and FID.

**Strengths:**

1. Provides some insight by exploring a new VAE decoding method.
2. Theoretical derivation makes sense and enhances the interpretability of the approach.

**Weaknesses:**

1. A minor point: In the abstract, you mention, "We evaluate our approach by assessing both reconstruction (rFID) and generation quality (FID)," but why is FID used to validate generation quality? I see you used IS in the experiment section instead.

2. The novelty is limited. The core innovation is only improving the VAE decoder, which offers minimal technical contribution.

3. The Introduction mentions various tokenizers, but the actual comparison seems to focus solely on VAE. What about discrete tokenizers like VQVAE and VQGAN?

4. You mention that the proposed model’s inference time is better than VAE, but also admit that "it requires more compute costs than VAE due to its U-Net design." In Section 5’s Discussion, you briefly touch on potential future optimizations. Does this imply that this limitation in your model is unsolvable?

5. You discuss VAE optimization in the context of Diffusion Models, but the inference time you’re referring to compares VAE’s reconstruction speed. In practice, the major time consumption in generation models isn’t in the VAE part. The faster inference time you mention is insignificant because the time it saves is negligible within 50 DDIM steps.

Based on my points above, I believe this paper does not meet the acceptance standards of ICLR, as it lacks innovation, delivers average results, and has insufficient technical contribution.

**Questions:**

See the weaknesses above.

---

> ### Author Response · Authors · 2024-11-22
> **Response to Reviewer VC75**
>
> Thank you for your valuable comments. We believe there are misunderstandings of our method in Q1, Q4, and Q5. We would like to address all your concerns below. Please let us know if you have any further questions.
>
> ***Q1:  Why is FID used to validate generation quality? I see you used IS in the experiment section instead.***
>
> FID is a standard metric to measure image generation quality which is widely used in the literature. We reported both FID and IS (together with Precision and Recall) in Table 2 of the main paper for evaluating image generation quality.
>
> ***Q2: The novelty is limited. The core innovation is only improving the VAE decoder, which offers minimal technical contribution.***
>
> Please refer to Q1 in the common response for justifications on our novelty and contributions.
>
> ***Q3: The Introduction mentions various tokenizers, but the actual comparison seems to focus solely on VAE. What about discrete tokenizers like VQVAE and VQGAN?***
>
> We provide a comparison with VQ models in the table below. Since the main focus of our paper is on improving VAEs for latent diffusion models, which require continuous latents, implementing VQ versions of our model is left for future work. We believe the techniques proposed in our paper can be easily transferred to VQ models, as our method does not rely on specific assumptions about the encoder outputs.
>
> | Models | Latent dim. | Vocabulary size | ImageNet rFID-50K | COCO2017 rFID-5K |
> | :---------------- | :------: | :------: | :------: | :------: |
> | VQGAN | 4 | 1024 | 1.44 | 6.58 |
> | ViT-VQGAN | 32 | 8192 | 1.28 | - |
> | LlamaGen | 8 | 16384 | 0.59 | 4.19 |
> | e-VAE (H) | 4 | - | 0.38 | 3.65 |
>
> ***Q4: You mention that the proposed model’s inference time is better than VAE, but also admit that "it requires more compute costs than VAE due to its U-Net design." In Section 5’s Discussion, you briefly touch on potential future optimizations. Does this imply that this limitation in your model is unsolvable?***
>
> Thank you for your question. To clarify, we have not claimed in the paper that the proposed model’s inference time is better than that of VAEs. As noted in L357, we acknowledge that our model has lower throughput compared to traditional VAEs due to the UNet-based diffusion decoder. However, we do not consider this limitation "unsolvable." In Section 5, we outline potential future optimizations, such as adopting lightweight architectures or implementing a patch-based decoder, which we believe can significantly reduce the compute cost while retaining the model's performance. These approaches are promising directions for addressing this limitation, and we are optimistic about their feasibility.
>
> ***Q5: You discuss VAE optimization in the context of Diffusion Models, but the inference time you’re referring to compares VAE’s reconstruction speed. In practice, the major time consumption in generation models isn’t in the VAE part. The faster inference time you mention is insignificant because the time it saves is negligible within 50 DDIM steps.***
>
> The latent diffusion model (LDM) remains consistent across all experiments, resulting in the constant inference time for the LDM. Consequently, the decoding time of the VAEs become a key factor influencing the overall inference speed. This is why our analysis in the main paper focuses on comparing the decoding speeds of different VAEs.
>
> On the other hand, our e-VAE supports a high compression rate, reducing the token length to one-fourth (x1/4) while preserving as much information as the original token length. This reduction can significantly accelerate latent diffusion model generation, decreasing overall inference time while maintaining favorable generation quality (please refer to Q2 in the common response).

---

> > ### Comment · Reviewer_VC75 · 2024-11-24
> >
> > 1. I question your statement that "the decoding time of the VAEs becomes a key factor influencing the overall inference speed." This is counterintuitive and lacks experimental evidence. Anyone who has run related code knows that the encoding and decoding processes themselves are not particularly time-consuming, while the denoising process of the Diffusion UNet accounts for the majority of the inference time.
> >
> > 2. Your mention of "outline potential future optimizations" inherently suggests that the issue is unsolvable within the current framework, doesn’t it?
> >
> > The authors’ arguments are not particularly convincing, and I am inclined to stick with my original score.

---

> ### Author Response · Authors · 2024-11-26
> **Thank you, we would like to make further clarification**
>
> We thank you for your reply, and would like to further clarify our claims.
>
> Q1. To clarify our statement, it's important to note that our analysis focuses on the impact of different VAEs on inference time **while keeping the LDM unchanged**. This is because our comparisons (Tables 1 and 2 in the main paper) utilize the same LDM throughout, ensuring its inference time remains unchanged. In this controlled setting, the decoding time of the VAE becomes the remaining factor influencing variations in overall inference time.
>
> While we acknowledge that the LDM generally dominates overall inference time, our statement aims to highlight the significant role of VAE decoding time within the above specific context. To make this clear, we propose rephrasing our statement to: **“When utilizing a fixed LDM, the decoding time of the VAE emerges as the remaining primary factor affecting overall inference time.”**
>
> Q2. We respectfully argue that our statement does not inherently suggest that the issue is unsolvable within the current framework. **Our statement was intended to highlight an area for future improvement, rather than suggest a fundamental limitation.**
> To further clarify, we note that (1) our proposed optimizations based on model architecture and decoding method are complementary to the current framework; (2) since our decoder utilizes a diffusion model, any advancements in diffusion model efficiency will directly benefit our framework and contribute to addressing the issue.
>
> We will revise our statement to better reflect this and avoid any misunderstanding.

---

> > ### Author Response · Authors · 2024-11-27
> >
> > Dear Reviewer VC75,
> >
> > We appreciate your valuable feedback. In our last response, we have provided further clarifications on our statement on inference time and potential future optimizations.
> >
> > We kindly inquire whether these clarifications have adequately addressed your concerns. Please feel free to let us know if you have any further questions.
> >
> > Thank you very much!
> >
> > Best regards,
> >
> > Authors of Paper 332

---

### Official Review · Reviewer_xmqC · 2024-10-27

**Soundness:** 4
**Presentation:** 3
**Contribution:** 1
**Rating:** 5
**Confidence:** 4

**Summary:**

This paper proposes using a diffusion-based decoder for visual tokenization, replacing traditional single-step decoding with iterative refinement. The authors compare U-Net and DiT diffusion architectures, finding U-Net to be superior. Compared to VQGAN, this approach also achieves better reconstruction (rFID) across all scales. Finally, the authors assess image generation quality and perform an ablation study.

**Strengths:**

- Uses a flow-based model, which could serve as a major upgrade over the previous diffusion-based decoder.
- Adds adversarial training to the pipeline, potentially retaining the benefits of GANs compared to VQGAN, and allowing for a reduction in sampling steps.
- Includes both U-Net and DiT architectures with comparisons across multiple model sizes, making this an extensive experiment.
- It’s beneficial to include image generation quality, though this aspect might not be the paper’s main focus; it’s expected that image generation quality should align with image reconstruction quality.

**Weaknesses:**

- The authors did not use DiT with a patch size of 2, which is considered optimal in the DiT paper and is the most widely used; however, the computational limitations are understandable.
- Image reconstruction evaluation relies heavily on a single metric, rFID (across both Tables 1 and 2). While it’s reasonable for authors to report preferred metrics, it would strengthen the paper to include additional commonly used metrics, such as PSNR, SSIM, and LPIPS, to provide a more comprehensive assessment.
- The paper lacks mention and discussion of previous diffusion-based autoencoders like DiffusionAE (https://diff-ae.github.io/) or DiVAE (https://arxiv.org/abs/2206.00386).
- Despite testing a new flow-based model and architecture, the conclusions on diffusion-based decoding do not significantly expand on those from previous studies.
- Diffusion-based decoders have been shown to improve upon standard VQGAN-based decoders (since 2022) and have seen applications like in 4M-21  (https://4m.epfl.ch). However, the reason that it not become popular, i believe, is time-intensive both training and inference, a challenge that this paper does not seem to overcome yet.

**Questions:**

- What are the "new insights" you mention in the abstract?
- "by" in line 216

---

> ### Author Response · Authors · 2024-11-22
> **Response to Reviewer xmqC**
>
> Thank you for your valuable comments. Below we provide a point-by-point response to all of your questions. Please let us know if you have any further questions.
>
> ***Q1: Why not using DiT with a patch size of 2?***
>
> In our preliminary experiments, we explored the DiT/2 but found it unsuitable for high-resolution image decoding. For example,  the Transformer's computational cost increases quadratically with token length, making it prohibitively expensive to decode a 256x256 image, which results in 128^2 tokens. Notably, the original DiT’s optimal patch size of 2 was designed for latent-level generation, where token lengths are typically small and manageable. In contrast, our decoder operates at the pixel level, leading us to consider only patch sizes of 4 and 8 for DiT in our experiments. Ultimately, we found ADM to produce better results and to be more suitable for this task.
>
> ***Q2: It would strengthen the paper to include additional commonly used metrics, such as PSNR, SSIM, and LPIPS, to provide a more comprehensive assessment.***
>
> Under the Stable Diffusion configuration, e-VAE achieves 0.152 LPIPS, 25.11 PSNR, and 0.71 SSIM on ImageNet, performing comparably to the standard SD-VAE. As discussed in Section 5 of the main paper, our approach prioritizes preserving the overall perceptual distribution of images rather than achieving pixel-perfect reconstruction. This aligns with our focus on perception-based compression under high compression rates. As a result, e-VAE excels in metrics such as rFID, which capture differences in perceived image distributions, rather than in pixel-level metrics like PSNR and SSIM.
>
> ***Q3: The paper lacks mention and discussion of previous diffusion-based autoencoders like DiffusionAE or DiVAE.***
>
> Please refer to Q1 in the common response for discussions on DiffusionAE and DiVAE. We also add discussions with these works in the paper revision.
>
> ***Q4: Clarification on our contributions. What are the "new insights" you mention in the abstract?***
>
> Please refer to Q1 in the common response for justifications on our contributions and impact.
>
> ***Q5: However, the reason that it not become popular, i believe, is time-intensive both training and inference, a challenge that this paper does not seem to overcome yet.***
>
> Our approach tackles the inefficacy of diffusion decoders by enhancing both training and inference speed. Unlike traditional diffusion models, e-VAEs could be trained on low-resolution images and well-generalized to high-resolution images (see Table 1). For example, our model achieves a training speed of 9 steps/sec on 128x128 images, comparable to VAEs (11 steps/sec). During inference, our approach can achieve fast decoding (1–3 steps) through the integration of our proposed objectives and parameterizations (Please see Q1 in the common response for detailed justifications). We believe further efficiency gains are anticipated with improved architectures and decoding methods, as discussed in Section 5.
>
> ***Q6: Comment on the writing.***
>
> Thank you for your meticulous review. We will fix the typo in the paper revision.

---

> > ### Author Response · Authors · 2024-11-27
> >
> > Dear Reviewer xmqC,
> >
> > We sincerely appreciate your valuable feedback and suggestions. In our rebuttal, we have provided explanations regarding the DiT/2 results and experimental comparisons on pixel-level metrics, our contributions and differences from the previous works you pointed out, and our improvements on model efficiency.
> >
> > We kindly inquire whether these results and clarifications have adequately addressed your concerns. Please feel free to let us know if you have any further questions.
> >
> > Thank you very much!
> >
> > Best regards,
> >
> > Authors of Paper 332

---

### Official Review · Reviewer_CU6D · 2024-10-28

**Soundness:** 3
**Presentation:** 3
**Contribution:** 2
**Rating:** 5
**Confidence:** 4

**Summary:**

The paper introduces $\epsilon$-VAE, which replaces the decoder with a UNet diffusion model. This work focuses on tokenization for latent diffusion models and substitutes the deterministic decoder with a diffusion process, aiming to achieve higher compression rates and improved reconstruction quality, thereby enhancing the generation quality of downstream generative models. Experiments are conducted on ImageNet using evaluation metrics such as FID, rFID, IS, Precision, and Recall.

**Strengths:**

The proposed $\textit{denoising as decoding}$ is interesting

Performance significantly exceeds the VAE

**Weaknesses:**

1. The approach involves using a diffusion model instead of a decoder. However, vanilla diffusion operates in latent space, resulting in generated latent code that do not match the image size. Without an additional decoder, how can images of the correct size be generated?

2. A major limitation of the vanilla diffusion model is its multi-step iteration, which results in longer inference times. Many existing methods have addressed this issue by reducing the number of inference steps (e.g., LCM, SD-Turbo, SwiftBrush). The decoder of the $\epsilon$-VAE also requires multiple inference steps, limiting its scalability to more general tasks (such as text-to-image tasks and downstream applications) and increasing inference time. Therefore, a comparison between a one-step $\epsilon$-VAE and a traditional VAE should be provided.

SwiftBrush: One-Step Text-to-Image Diffusion Model with Variational Score Distillation (CVPR'24)

SD-Turbo: Adversarial Diffusion Distillation

LCM: Latent Consistency Models: Synthesizing High-Resolution Images with Few-Step Inference

**Questions:**

The $\epsilon$-VAE transforms the standard decoding process of a VAE into an iterative denoising task. However, this iterative process introduces additional inference time, which is not reported in the paper.

---

> ### Author Response · Authors · 2024-11-22
> **Response to Reviewer CU6D**
>
> Thank you for your constructive comments. Below we provide a point-by-point response to all of your questions. Please let us know if you have any further questions.
>
> ***Q1: Aligning latents and image resolutions.***
>
> As described in L197–199 and illustrated in Figure 1, low-resolution latents obtained from the encoder are upsampled to match the resolution of the noisy image before being passed into the diffusion decoder. We also explored an alternative conditioning strategy that modulates the decoder layers through AdaGN (in Appendix C). However, due to its increased computational complexity, we chose to use resizing and concatenation for their simplicity and effectiveness.
>
> ***Q2: Sampling efficiency.***
>
> Please refer to Q1 in the common response. To further emphasize this aspect, as shown in Table 3 and Figure 3 (left), our e-VAE achieves competitive performance with single-step decoding and delivers the best results with three steps, all without relying on distillation or consistency regularizations. This is made possible by our parameterization strategy (velocity-prediction) and the integration of LPIPS (applied to $\hat{x}_0$) and GAN losses (trajectory matching). Such sampling efficiency has not been demonstrated by any prior diffusion decoder-based VAE approaches.
>
> ***Q3: Throughput.***
>
> As noted in L357, we also measure the actual throughput. Admittedly, our framework has lower throughput compared to traditional VAEs due to the use of a UNet-based diffusion decoder. However, we believe this limitation can be mitigated in the future by adopting lightweight architectures or implementing patch-based decoder, as described in Section 5.

---

> > ### Comment · Reviewer_CU6D · 2024-11-25
> > **Restatement for Weakness2**
> >
> > The author has not addressed my concerns. Therefore, I would like to restate Weakness 2 and kindly request a response.
> >
> > I remain puzzled by the trade-off between simply improving the generation quality of the VAE and the resulting increase in inference time. Specifically:
> >
> > 1. The VAE enables diffusion-based models to train and infer in the latent space, thereby improving efficiency in both processes. Examples include Stable Diffusion and DiT. However, the main challenge for vanilla diffusion-based models is inference time, which methods like SwiftBrush address by distilling into few-step models. If $\epsilon$-VAE attempts to use a diffusion model as a decoder, could replacing the VAE in multi-step diffusion models with $\epsilon$-VAE reduce the original diffusion model’s inference steps, thereby shortening inference time without compromising generation quality? Can the generative capability loss caused by reducing the number of steps be compensated by the decoder of $\epsilon$-VAE?
> >
> > 2. $\epsilon$-VAE is trained on ImageNet. If it cannot replace the VAE to achieve acceleration or improve generation quality, then its improvement in quality alone should be compared with other models trained on ImageNet, such as DiT, rather than solely with VAE (Table1&2).
> >
> > Moreover, if $\epsilon$-VAE is an enhanced version of VAE, it should be capable of performing all tasks that a VAE can accomplish.

---

> ### Author Response · Authors · 2024-11-26
> **Thank you, we would like to further address your concern on Weakness 2**
>
> Thank you for your reply and the opportunity to further clarify this aspect of our work.
>
> To address your concern, we want to emphasize that **the e-VAE is indeed designed as a better alternative to the standard VAE which are paired with latent diffusion models (LDMs) for image generation**. As demonstrated in Table 2 of the main paper, e-VAEs’ generation qualities consistently outperform VAEs in this context. These comparisons were conducted by first denoising in the latent space using an LDM (with a fixed architecture, i.e., DiT-XL/2) and then decoding the resulting latents into images using decoders from either VAEs or e-VAEs.
>
> To better illustrate the trade-off between generation quality and inference time when paired with LDMs, we provided the following table (reorganized from the second table in "Common response to all reviewers [Part 2]") with  'f8' and 'f16' denoting downsampling rates of 8 and 16 for the VAEs/e-VAEs, respectively.
>
> | Models | LDM token length | FID-50K | Inference time (s) |
> | :------- | :-------: | :-------: | :-------: |
> | (a) SD-VAE (f8) + DiT-XL/2 | 32x32 | 9.42 | 35.19 |
> | (b) e-VAE (f8) + DiT-XL/2 | 32x32 | 9.39 | 36.88 |
> | (c) e-VAE (f16) + DiT-XL/2 | 16x16 | 10.68 | 10.42 |
>
> This table highlights two key observations regarding the trade-off:
>
> - (a) v.s. (b): **Replacing the SD-VAE (f8) directly with e-VAE (f8) results in better generation quality but slightly longer inference time.** This is attributed to the UNet-based diffusion decoder in the e-VAE, which improves quality while adding computational complexity. Our improvements are more significant when the setting is challenging, e.g., with higher downsampling rates and weaker encoder (see Table 2 of the main paper).
>
> - (a) v.s. (c): **Replacing the SD-VAE (f8) with e-VAE (f16) results in slightly worse generation quality but significantly reduced inference time.** This reduction is achieved by ***decreasing the input token length to the LDM***, rather than reducing the LDM's inference steps, as we maintain the original LDM design unchanged throughout our experiments. This is supported by the inherent ability of e-VAEs to provide significantly better reconstruction quality than VAEs when the downsampling rate is high (see Table 1 of the main paper).
>
> It's important to note that our method is orthogonal to other techniques aimed at improving LDM inference time through distillation. These techniques, as mentioned in the review, could be directly applied to our framework since we do not modify the LDMs themselves.
>
> We hope the above explanation addresses your concern. Please let us know if you have any further questions.

---

> ### Author Response · Authors · 2024-11-27
>
> Dear Reviewer CU6D,
>
> We sincerely appreciate your valuable feedback on our paper. In our last response, we have provided detailed explanations on the setup of our model for image generation and trade-off between generation quality and inference time.
>
> We kindly inquire whether these clarifications have adequately addressed your concerns. Please feel free to let us know if you have any further questions.
>
> Thank you a lot!
>
> Best regards,
>
> Authors of Paper 332

---

> ### Comment · Reviewer_CU6D · 2024-11-29
>
> Improving the quality of generated images using $\epsilon$-VAE comes at the cost of increased inference time. Similarly, increasing the number of sampling steps can also enhance the quality of the generated images. Why use $\epsilon$-VAE instead of increasing the number of inference steps in the vanilla SD?.
>
> The authors mention that "our method is orthogonal to other techniques aimed at improving LDM inference time through distillation." However, I have concerns about this claim. It is necessary to validate this through experiments by combining existing distillation models, such as SD-Turbo, LCM, LCM-LoRA, and SwiftBrush.
>
> SD-Turbo: Adversarial Diffusion Distillation \
> LCM:  Latent Consistency Models: Synthesizing High-Resolution Images with Few-Step Inference. \
> LCM-LoRA: LCM-LoRA: A Universal Stable-Diffusion Acceleration Module \
> SwiftBrush: One-Step Text-to-Image Diffusion Model with Variational Score Distillation

---

> > ### Author Response · Authors · 2024-12-02
> >
> > Thanks for your reply. We provided our response to your questions below.
> >
> > ***Q1. Why use e-VAE instead of increasing the number of inference steps in the vanilla SD?***
> >
> > This is because the inference cost of the latent diffusion model (i.e., DiT-XL/2) significantly exceeds that of e-VAE. Specifically, in our experiment, we observe that each inference step of the latent diffusion model was approximately 10 times more computationally expensive than that of e-VAE. Therefore, utilizing e-VAE to improve the generation quality offers greater efficiency gains compared to increasing the number of inference steps in the vanilla latent diffusion model.
> >
> > ***Q2. Combining existing distillation models.***
> >
> > Thank you for the suggestion to validate combining existing distillation models with e-VAE. While intriguing, conducting such extensive experiments within the limited rebuttal period and provided guideline is unfortunately infeasible. On the other hand, we wish to emphasize that (1) our statement is considered as a reasonable assumption in previous works (e.g., [1]) on improving the VAE model while keeping the latent diffusion model unchanged; (2) investigating distillation models falls outside the scope of the current paper, as our primary goal is to improve the VAE itself. We will certainly prioritize this valuable feedback in our future work.
> >
> > [1] “LiteVAE: Lightweight and Efficient Variational Autoencoders for Latent Diffusion Models”, NeurIPS 2024.

---

> > > ### Comment · Reviewer_CU6D · 2024-12-03
> > >
> > > Thank you for your detailed rebuttal and the additional experimental results. However, after carefully reviewing your response, my concerns regarding the contribution of the work remain unresolved.
> > >
> > > This study appears to primarily integrate existing technologies, such as VAE and diffusion models. While it shows some improvements over vanilla VAE, this comes at the cost of increased inference time.
> > >
> > > Given these considerations, I believe it is appropriate to maintain the original rating.

---

### Official Review · Reviewer_bhvX · 2024-10-31

**Soundness:** 3
**Presentation:** 3
**Contribution:** 4
**Rating:** 8
**Confidence:** 4

**Summary:**

This paper proposes a new generative model, named $\epsilon$-VAE, combining together techniques from Variational Autoencoders (VAEs), Generative Adversarial Networks (GANs) and Diffusion Models (DM), to accomplish both image reconstruction tasks (super-resolution), and generation. In particular, the architecture of $\epsilon$-VAE consists of a decoder in the style of a Variational Encoder, mapping the input image to its latent representation, and a conditional DM-based decoder, which maps the latent representation back to the image domain. The experimental section is mostly well-done, and it clearly shows that -VAE beats modern VAE model in basically every analyzed setup.

**Strengths:**

- **Originality**: The idea presented in the paper is original and, in my opinion, non trivial.
- **Quality**: Despite a few observations, the overall quality of the paper is great.
- **Clarity**: The paper is well-written and the results clearly presented.
- **Significance**: The results from the presented experiments prove that the idea is appealing to the scientific community. While being relatively simple, the idea presented in this work is worth to be published.

**Weaknesses:**

There are a few points that needs to be clarified for the paper to be accepted.

-	First of all, the notation is confusing. The authors use the terms “tokenization” and “pre-processing” to indicate what is simply a CNN-based encoder, in the style of any image autoencoder network. While this is in general not a big issue, I suggest to clarify this aspect more clearly at the beginning of the paper, since the term “tokenization” is usually associated with language models, while “encoder” is more common in image processing field.
-	Secondly, I do not understand why the name “$\epsilon$-VAE” contains a clear reference to Variational Autoencoders, while after the modifications provided by the authors the resulting model is far from being a VAE. For example, the primary difference between VAE and any other Autoencoder is the training loss (i.e. the ELBO objective) and the presence of an approximate probability distribution of the latent space, in the form of q(z | x). None of these two properties appear to be present in the proposed model, which looks closer to a Diffusion Model conditioned on latent representation obtain through a convolutional encoder, than to a VAE. I understand that this is close to what happens in VQ-VAE and VQ-GAN (Esser et. al, 2021), but in this case the absence of an explicit fit of the prior distribution during training is justified by the assumption of a discrete latent space, which avoids the gradient to be backpropagated through the network, while in your setup the latent variable is continuous.
-	A consequence of the previous point is that in the experimental section you compared your result with a moden VAE model (from Esser et al, 2021). While the compared model is close to be state-of-the-art in the field of Variational Autoencoder, its performance as a generative model are easily surpassed by state-of-the-art Diffusion Models. Since I believe $\epsilon$-VAE is a Diffusion Model, it would be interesting to see results compared against a Diffusion Model, instead of against a Variational Autoencoder. This observation is also supported by Figure 2 (left), where ADM by itself clearly reaches lower rFID than the VAE model to which you compared with. Thus, I suggest the authors to compare $\epsilon$-VAE with ADM in the experiments.
-	Lastly, for the big part of the paper, the authors did not declare the number of diffusion steps employed by $\epsilon$-VAE. Based on the Iterative and Stochastic Decoding paragraph at the end of the paper, I infer that you maybe used either 1 or 3 diffusion steps. While a low number of steps is of common use in Diffusion Models applied to solve Image Reconstruction tasks, one could argue that a single-step Diffusion Model is not really a Diffusion Model, but a simple application to a conditional UNet denoiser.

**Minor Comments.**

1.	In line 32, “Tokenization is a essential in both”. Remove the “a”.
2.	In line 105, I believe that in the definition of sigma_t, it should be an “alpha” instead of an “a”.
3.	In the “Impact of proposal” section, the notation (1), (2), … is confusing because it looks like it refers to equations. I suggest to instead use some variants such as (P1), (P2), …, to indicate “Proposal 1”, “Proposal 2”, ….

**Questions:**

I included a few questions on the "Weakness" section.

---

> ### Author Response · Authors · 2024-11-22
> **Response to Reviewer bhvX**
>
> Thank you for your valuable comments. Below we provide a point-by-point response to all of your questions. Please let us know if you have any further questions.
>
> ***Q1: Clarification on the term "tokenization”.***
>
> As noted in L33, tokenization refers to the process of transforming data into compact representations, which can be either continuous or discrete. As suggested, we will further clarify that continuous latents are typically obtained through an encoder in vision tasks, while discrete tokens are usually derived using embeddings in language tasks.
>
> ***Q2: Clarification on the naming “e-VAE”.***
>
> Thanks for pointing this out. We use the term “e-VAE” because the parameterizations in the diffusion decoder, whether epsilon-prediction or velocity-prediction, essentially transform into a traditional score-matching objective (L135–137), which can be interpreted as a reweighted version of the ELBO. Furthermore, our proposal directly builds upon the original VAE framework by replacing the reconstruction objective with the score-matching objective, while retaining LPIPS and GAN losses by ensuring compatibility with the diffusion decoder. For these reasons, we believe the term "e-VAE" is appropriate. However, if this naming remains unclear, we are open to considering alternative terms.
>
> ***Q3: Comparison with plain diffusion ADM.***
>
> As suggested, under the same training setup, we directly trained a plain diffusion model (ADM) for comparison, which resulted in rFID score of 38.26. Its conditional form is already provided as a baseline in Table3, achieving 28.22. This demonstrates that our conditional form $p(x_{t-1}|x_t,z)$ offers a better approximation of the true posterior $q(x_{t-1}|x_t,x_0)$ compared to the standard form $p(x_{t-1}|x_t)$. By further combining LPIPS and GAN loss, we achieve rFID of 8.24, outperforming its VAE counterpart, which achieves 11.15. With better training configurations, our final rFID improves to 6.24. This progression, from plain diffusion ADM to e-VAE, underscores the significance of our proposals and their impact.
>
>
> ***Q4: Sampling steps.***
>
> We apologize for the unclear description. As noted in Table 3 and Figure 3 (left), our e-VAE achieves the best results using three sampling steps, while still delivering competitive performance even with a single step. We will ensure this is clarified more explicitly in the text.
>
> ***Q5: Typos.***
>
> Thank you for your meticulous review. We will make the suggested corrections in the manuscript.

---

> > ### Comment · Reviewer_bhvX · 2024-11-24
> >
> > Thank you for replying to all my concerns. I am still not completely convinced by the name "epsilon-VAE" because it makes readers think they are going to read about a variant of Variational Autoencoders, while they will find a diffusion model applied to the latent space of an Autoencoder. But, clearly, the name is just a preference.
> > Everything is clear for me, thanks!

---

> > > ### Author Response · Authors · 2024-11-25
> > > **Thank you**
> > >
> > > Thanks for your reply. We are glad our responses have clarified your questions. To ensure consistency and avoid any potential confusion during the rebuttal period, we would prefer to retain the current method name for now. However, we highly value your suggestion and will certainly give it careful consideration in the following revision.

---

> > > > ### Comment · Reviewer_bhvX · 2024-11-26
> > > > **Agreement**
> > > >
> > > > Yes, I completely agree with not changing the name at this point as it would require reformulating the whole content. I confirm my Rating as it was already pretty high, and it is mainly influenced by the content of the work (which I believe is interesting) rather than the minor comment I had on the exposition.

---

> > > > > ### Author Response · Authors · 2024-11-27
> > > > >
> > > > > Dear Reviewer bhvX,
> > > > >
> > > > > Thank you again for your understanding and prompt feedback!
> > > > >
> > > > > Best regards,
> > > > >
> > > > > Authors of Paper 332

---

### Official Review · Reviewer_drdo · 2024-11-03

**Soundness:** 2
**Presentation:** 3
**Contribution:** 2
**Rating:** 5
**Confidence:** 4

**Summary:**

This paper proposes ε-VAE as a new tokenizer to replace traditional VAE, specifically implementing a diffusion model as the decoder process instead of the original VAE decoder.

Advantages:
1. The idea is straightforward and appears promising
2. The paper is well-written and easy to understand, I can follow all equations in the paper. There is no innovation in the mathematical aspect.

Disadvantages:
1. The novelty is limited
2. The quantitative metrics are subpar
3. The reconstruction visual quality is inadequate

According to LLamaGen [1], on the 256×256 ImageNet 50k validation set, SD-VAE achieves rFID 0.820 and SDXL-VAE obtains an rFID of 0.68. Similar results can be found in [5]. However, this paper's performance is significantly lower than the original VAE results.

Based on my reconstruction experience, after carefully examining the figures presented in the paper, the visual quality of the reconstructions is not satisfactory.
Please show some results that can reconstruct face details and text details.

I have some suggestions,
1. Test the method on COCO dataset
2. Further improve current results, as there is still a considerable gap to SOTA performance on ImageNet
3. Evaluate reconstruction performance on video data

References:
[1] LLamaGen
[2] VAR
[3] MAR
[4] SDXL
[5] https://github.com/LTH14/mar/issues/3
[6] MagVit2

**Strengths:**

Advantages:
1. The idea is straightforward and appears promising
2. The paper is well-written and easy to understand, I can follow all equations in the paper. There is no innovation in the mathematical aspect.

**Weaknesses:**

Disadvantages:
1. The novelty is limited
2. The quantitative metrics are subpar
3. The reconstruction visual quality is inadequate

According to LLamaGen [1], on the 256×256 ImageNet 50k validation set, SD-VAE achieves rFID 0.820 and SDXL-VAE obtains an rFID of 0.68. Similar results can be found in [5]. However, this paper's performance is significantly lower than the original VAE results.

Based on my reconstruction experience, after carefully examining the figures presented in the paper, the visual quality of the reconstructions is not satisfactory.
Please show some results that can reconstruct face details and text details.

I have some suggestions,
1. Test the method on COCO dataset
2. Further improve current results, as there is still a considerable gap to SOTA performance on ImageNet
3. Evaluate reconstruction performance on video data

References:
[1] LLamaGen
[2] VAR
[3] MAR
[4] SDXL
[5] https://github.com/LTH14/mar/issues/3
[6] MagVit2

**Questions:**

see above

---

> ### Author Response · Authors · 2024-11-22
> **Response to Reviewer drdo**
>
> Thanks for your insightful comments. Most of your questions are answered in the common response, and we provide a summary below. We sincerely hope you could reconsider your rating regarding our newly added results. Please let us know if you have any further questions.
>
> ***Q1: Novelty.***
>
> Please refer to Q1 in the common response for justifications on our novelty and contributions.
>
> ***Q2: Additional quantitative results on ImageNet and COCO.***
>
> Please refer to Q2 in the common response for the additional quantitative results that aligns with the SD-VAE configuration. We also included results on COCO as suggested. We leave reconstruction on video data as our future work since (1) the major focus of this paper is image reconstruction; (2) the rebuttal time and compute budget are limited.
>
> ***Q3: Additional visual results for reconstruction.***
>
> Please refer to Q3 in the common response for the additional visual results on face details and text details. Overall, we find that our models perform significantly better than SD-VAE when the model and training configurations are matched.

---

> > ### Comment · Reviewer_drdo · 2024-11-25
> >
> > Thank the authors for the rebuttal. I am partly satisfied with the experiments added by the authors, but I still insist that the novelty of this paper is limited especially after reading comments from the other reviewers.  I will raise my score to 5 because of the added experiments.

---

> > > ### Author Response · Authors · 2024-11-25
> > > **Thank you for your reply**
> > >
> > > Thank you for your thoughtful reply and for reconsidering your rating. We really appreciate your detailed suggestions regarding our experiments which improve our paper.
> > >
> > > To help us further enhance the clarity and impact of our work, could you please elaborate on your concerns about the novelty of our approach? Understanding your perspective will enable us to better address this aspect and ensure the paper effectively communicates our contributions.

---

> > > > ### Author Response · Authors · 2024-11-27
> > > >
> > > > Dear Reviewer drdo,
> > > >
> > > > We sincerely appreciate your valuable feedback and reconsideration on our paper. Could you please further elaborate on your concerns about our novelty to let us better address this aspect? Thank you very much!
> > > >
> > > > Best regards,
> > > >
> > > > Authors of Paper 332

---

### Official Review · Reviewer_9E74 · 2024-11-04

**Soundness:** 4
**Presentation:** 4
**Contribution:** 4
**Rating:** 8
**Confidence:** 5

**Summary:**

This paper proposes an autoencoder trained with diffusion loss, together with LPIPS loss and GAN loss applied on the estimated sample from the diffusion decoder. The authors show improved reconstruction and generation quality comparing to the prior GAN-based autoencoders, which demonstrates the effectiveness of the diffusion loss in joint training.

**Strengths:**

**Update after rebuttal:**

```
The authors have well addressed my questions and I will keep my rating.

I also looked on the remaining concerns from other reviewers, and do not feel they are major weaknesses:

1. Novelty: To the best of my knowledge, and according to the papers other reviewers listed, I did not find any published paper shows the same results that joint training AE with diffusion loss helps rFID and visual quality, which is an important result to understand the strength of diffusion autoencoders.

2. Inference time: To me this is not the key focus. Any method with diffusion models uses iterative sampling. As long as all reviewers agree on the improvement of evaluation metric and visual quality, this is not a major weakness. Especially given the recent advancements of faster samplers and one-step diffusion distillation.

```

---

1. Diffusion loss for autoencoder training is an important direction to explore. This work is one of the first works that show promising results.
2. The proposed method outperforms prior GAN-based autoencoder on ImageNet with the common metric FID. The trend of compression ratio also shows the advantage of the diffusion loss.
3. The evaluation is comprehensive and well-organized. The number of trainable parameters are also listed for better comparison.

**Weaknesses:**

1. The LPIPS and GAN loss are applied on the estimated sample, which seems to be not accurate that may cause objective bias in theory.
2. It is not very easy to note the difference between the baseline VAE and the proposed eps-VAE in Figure 4 (images are compressed in the paper?), especially for 8x downsampling. A higher quality / higher resolution demonstration, zoom-in crops, or even selection on samples, could be helpful as visual comparison.
3. The generation FIDs are a bit high in Table 2 (though it could be due to the computation budget and could be a fair comparison with the same number of iterations).
4. A few related works [1, 2] that might be missing in the paper.

[1] Diffusion Autoencoders: Toward a Meaningful and Decodable Representation, CVPR 2022

[2] Würstchen: An Efficient Architecture for Large-Scale Text-to-Image Diffusion Models, ICLR 2024

**Questions:**

How is the baseline VAE (GAN-based autoencoder) designed and scaled up in the paper? Are they all re-trained to match the setting of eps-VAE? Is the number of channels 8 for Figure 4 and 5?

In particular, is there any quantitative results / visual samples correspond to downsampling-factor-8 and 4-channels setting for the baseline VAE, which is the default setting used in LDM / Stable Diffusion (for images at resolution 256 or 512)?

---

> ### Author Response · Authors · 2024-11-22
> **Response to Reviewer 9E74**
>
> Thanks for your positive feedback. Below we provide a point-by-point response to all of your questions. Please let us know if you have any further questions.
>
> ***Q1: The LPIPS and GAN loss are applied on the estimated sample, which seems to be not accurate that may cause objective bias in theory.***
>
> This is indeed a good question. Unlike traditional diffusion models, our e-VAEs utilize a diffusion model conditioned on the encoder outputs. This conditioning provides strong prior information, significantly improving the estimation quality of the samples (see Q3 from bhvX), thereby reducing potential objective bias.
>
> ***Q2: Visual comparisons at higher resolutions under the Stable Diffusion configuration.***
>
> Please refer to Q3 in the common response for visual comparisons between SD-VAE and e-VAE under x8 downsampling and 4 channels at resolutions of 256x256 and 512x512. Generally, e-VAE performs better when reconstructing complex regions such as human faces and small texts.
>
> ***Q3: The generation FIDs are a bit high in Table 2.***
>
> This is because we perform unconditional image generation, which usually results in higher FID compared to class-conditional image generation. Additionally, the employed VAEs have a fraction of parameters of their SD-VAE counterparts (6M vs. 34M) and use a higher compression rate (16 vs. 8), being trained under a more challenging configuration (see Q2 in the common response).
>
> ***Q4: A few related works [1, 2] that might be missing in the paper.***
>
> Thanks for pointing them out. Please refer to Q1 in the common response for our discussion. We also add discussions with these works in the paper revision.
>
> ***Q5: How is the baseline VAE (GAN-based autoencoder) designed and scaled up in the paper? Are they all re-trained to match the setting of eps-VAE? Is the number of channels 8 for Figure 4 and 5?***
>
> The design of baseline VAEs is summarized in L277 of the main paper, and they are scaled up in the same way as e-VAEs (see Table 4). Yes, all the baseline VAEs in the original paper are re-trained to match the setting of eps-VAE. The number of channels is 8 for Figures 4 and 5.

---

> ### Author Response · Authors · 2024-11-27
>
> Dear Reviewer 9E74,
>
> We sincerely appreciate your recognition and valuable feedback on our paper. In our response, we have provided detailed quantitative and high-resolution visual comparisons of our model against SD-VAE. Additionally, we have clarified the loss functions and baseline setups, and discussed the related work you pointed out.
>
> We kindly inquire whether these clarifications and results have adequately addressed your concerns. Please feel free to let us know if you have any further questions.
>
> Thank you a lot!
>
> Best regards,
>
> Authors of Paper 332

---

### Author Response · Authors · 2024-11-22
**Common response to all reviewers (Part 1)**

We thank all reviewers for their constructive feedback. We are encouraged that reviewers appreciate our work for: advancing autoencoders with diffusion loss as a promising direction (9E74, drdo), presenting an original and appealing idea (bhvX), offering interesting and insightful contributions (CU6D, VC75), achieving significant performance improvements over prior methods (9E74, CU6D), and providing extensive experiments (9E74, xmqC).

***Q1: Contribution.***

To clarify and emphasize our core contributions, we summarize our main findings, which distinguish our work from previous studies and address concerns regarding novelty (drdo, xmqC, VC75) and inference speed (CU6D, xmqC).

First, we thank 9E74 and xmqC for pointing out prior works ([1], [2], [3]) that have explored diffusion decoders conditioned on compressed latents of the input. Below, we outline the key differences between these works and e-VAE.

- __Synergizing Diffusion Loss with LPIPS and GAN Objectives:__ Previous works have not fully explored the synergy between diffusion decoders and standard VAE training objectives. In this work, we enhance state-of-the-art VAE objectives by replacing the reconstruction loss with a score matching loss and adapting LPIPS and GAN losses to ensure compatibility with the diffusion decoder. These modifications lead to significant improvements in autoencoding performance, demonstrated by lower rFID scores and faster inference.

- __Velocity Parameterization:__ We are the first to explore various parameterizations (e.g., epsilon and velocity) and show that modern velocity parameterization, along with the associated train and test-time noise scheduling, offers substantial benefits by greatly improving both reconstruction performance and sampling efficiency.

- __Single-step Decoding:__ Unlike previous diffusion-based decoders ([1], [2], [3]), which typically require ad-hoc techniques like distillation or consistency regularizations to accelerate inference ([4], [5], [6] noted by CU6D), our approach achieves fast decoding (1–3 steps) without such techniques. This is enabled by the integration of our proposed objectives and parameterizations (as shown in Table 3 and Figure 3-left).

- __Resolution Generalization:__ Last but not least, our e-VAE exhibits strong resolution generalization capabilities, a key property of standard VAEs. In contrast, models like DiffusionAE [1] and DiVAE [2] either lack this ability or are inherently limited. For example, DiVAE's bottleneck add/concat design restricts its capacity to generalize across resolutions.

We believe these contributions highlight the novelty and impact of our work. __While [1], [2], and [3] demonstrate the initial potential of diffusion decoders, we are the first to fully unlock their capabilities toward a more practical diffusion-based VAE, achieving strong rFID, high sampling efficiency, and robust resolution generalization.__

[1] Diffusion Autoencoders: Toward a Meaningful and Decodable Representation

[2] DiVAE: Photorealistic Images Synthesis with Denoising Diffusion Decoder

[3] Würstchen: An Efficient Architecture for Large-Scale Text-to-Image Diffusion Models

[4] SwiftBrush: One-Step Text-to-Image Diffusion Model with Variational Score Distillation

[5] SD-Turbo: Adversarial Diffusion Distillation

[6] LCM: Latent Consistency Models: Synthesizing High-Resolution Images with Few-Step Inference.

---

> ### Author Response · Authors · 2024-11-22
> **Common response to all reviewers (Part 2)**
>
> ***Q2: Additional quantitative results aligning with the Stable Diffusion setting.***
>
> In the main paper, we use a light-weight encoder (with 6M parameters), the compression rate of 16, and the channel dimension of 8 for 128x128 image reconstruction, since we focus on handling high compression rates. This configuration is more challenging than the traditional setup of VAEs in Stable Diffusion (SD-VAE), which leads to the performance gap.
>
> In this rebuttal, to address reviewers’ concerns on reconstruction quality (9E74, drdo), we provide additional results of our models under the same configuration with SD-VAE, i.e., with a standard encoder (with 34M parameters), the compression rate of 8, and the channel dimension of 4 for 256x256 image reconstruction. We report rFID on the full validation set of ImageNet and COCO2017. Comparisons with SD-VAE and SDXL-VAE are shown in the table below.
>
> | Models    | Decoder #params (M) | ImageNet rFID-50K | COCO2017 rFID-5K |
> | :-------- | :-------: | :-------: | :-------: |
> | SD-VAE  | 49.49  | 0.74 | 4.45 |
> | SDXL-VAE | 49.49  | 0.68 | 4.07 |
> | e-VAE (B)  | 20.63  | 0.52 | 4.24 |
> | e-VAE (M)  | 49.33  | 0.47 | 3.98 |
> | e-VAE (L)  | 88.98  | 0.45 | 3.92 |
> | e-VAE (XL)  | 140.63  | 0.43 | 3.80 |
> | e-VAE (H)  | 355.62  | 0.38 | 3.65 |
>
> We find that Epsilon-VAE outperforms SD-VAEs when the decoder sizes are similar, and our results could be further improved when we scale up the decoder.
>
> In addition, we want to highlight that when combined with latent diffusion models for class-conditional image generation, e-VAE is able to achieve comparable generation quality even when using only 25% of the typical token length required by SD-VAE. To show this, we train an additional e-VAE (M) under the Stable Diffusion configuration but double the downsampling rate. Then, we compare our models with SD-VAE by training DiT/2 under the class-conditional image generation setup (no classifier-free guidance) on ImageNet 256x256. We follow the setup in the DiT paper and all DiTs are trained with 1M steps. Results are shown in the table below.
>
> | VAE used in LDM    | VAE downsampling rate | LDM token length | ImageNet FID-50K |
> | :-------- | :-------: | :-------: | :-------: |
> | SD-VAE  | 8  | 32x32 | 9.42 |
> | e-VAE (M)  | 8  | 32x32 | 9.39 |
> | e-VAE (M)  | 16 | 16x16 | 10.68 |
>
> ***Q3: Additional high-resolution visual results aligning with the Stable Diffusion setting.***
>
> As requested by reviewers (9E74, drdo), in the appendix of the revised paper (Appendix D, Pages 21-22), we provide additional apple-to-apple visual comparisons between e-VAE and SD-VAE under the SD-VAE configuration at the resolutions of 256x256 and 512x512. We observe that e-VAE achieves significantly better visual qualities than SD-VAE when reconstructing local regions with complex textures or structures, such as human faces and small texts.

---

### Public Comment · ~Xu_Ma2 · 2025-02-07
**Thanks for the awesome work and would like to learn more**

Dear Authors,

Thank you for this excellent work!

I appreciate your contributions, especially using denoising as a decoder and optimizing it for better performance. I also find the idea of replacing single-step deterministic decoding with an iterative, stochastic denoising process very interesting. I agree that inference cost may not be a major issue, and traditional PSNR & SSIM evaluations might not be the best fit for diffusion-based decoders.

-----

I have a few questions and would love to hear your thoughts and learn more from the authors:

1. Have you considered fine-tuning a diffusion model as a decoder using latent representations as conditions, similar to SEED, EMU, and EMU2?

    [A] SEED:Planting a seed of vision in large language model

    [B] Emu: Generative Pretraining in Multimodality

    [C] EMU2: Generative Multimodal Models are In-Context Learners

2. What kind of latent features work best, low-level (pixel-level, VAE) or high-level (semantic, CLIP)? Would discrete representations be feasible?

3. Maybe autoregressive (AR) approach can better showcase the generation quality in Sec 4.2? Since Diffusion model and the proposed diffuison-based decoder might be little overlapped?

4. Would a log scale for parameters and channel dimensions in Fig. 2 better illustrate scaling?

5. Did you find diffusion-based decoders more robust than traditional VAE decoders? say more robust to the input latent, like error latent can also generate good-looking images?

Looking forward to your insights.  Thanks again for your great work!

------

Best,

Xu

---

> ### Public Comment · ~Long_Zhao2 · 2025-02-14
> **Thanks for your interest**
>
> Hey Xu,
>
> Thanks for your interest and kind words. We answer your questions below.
>
> 1. We feel these lines of work are somehow different from ours in the main goal. They focus on training stronger decoders to take more duties on the generation side in autoencoding, so that the latents could be used for modeling other perspectives (e.g., semantic information) potentially for specific purposes (e.g., unification); while, we remain on developing a VAE for achieving better visual qualities in pure generative models. I believe the direction you mentioned is indeed an important topic where our method could be extended to.
>
> 2. (1) Generally, it largely depends on the use case. In my own view, for a pure generative model, low-level latents are usually beneficial as they alleviate the duty of a decoder (low-level latents are easier for reconstruction); high-level/semantic latents are usually required in understanding models or unified models, while a heavier decoder (like the models in Q1) might be essential for good generation qualities in this case. (2) We expected that our method could work with discrete representations out-of-the-box, since we do not make any assumptions about the latent forms in our model.
>
> 3. Thank you for your suggestion. While we agree that incorporating AR approaches could enhance our current results, we believe that using diffusion models alone still provides a sufficient showcase. This is because latent prediction and autoencoding are two orthogonal functionalities, even diffusion models are applied in both cases.
>
> 4. Yes, this is indeed a good point. We will consider incorporating it in the revision.
>
> 5. While we haven't specifically tested such scenarios, traditional VAEs have demonstrated a certain degree of robustness to corrupted latents (https://arxiv.org/pdf/2409.16211), which we believe could be transferred to our method. Furthermore, our method's emphasis on perception-compression (see the “Discussion” section in our paper) might offer additional advantages in handling these cases.
>
> Thanks,
>
> Long

---

> > ### Public Comment · ~Xu_Ma2 · 2025-02-14
> >
> > Dear Dr. Long,
> >
> > Thanks a lot for those insightful thoughts, which inspire a lot! I also appreciate the valuable contributions in this work.
> >
> > Best,
> >
> > Xu

---

### Meta-Review · Area_Chair_XTrn · 2024-12-19

**Metareview:**

The submission presents an approach that integrates diffusion decoders into the VAE framework, but reviewers expressed concerns about its limited novelty, primarily focusing on replacing the VAE decoder without introducing substantial technical innovations. While the authors provided additional experiments and clarifications in their rebuttal, key issues like the practical trade-offs between quality improvements and increased inference time, as well as the lack of comparisons with state-of-the-art diffusion models or distillation techniques, remain inadequately addressed. Furthermore, the work's framing as a VAE enhancement was questioned due to deviations from standard VAE principles, and its contributions were perceived as incremental rather than groundbreaking.

**Additional Comments On Reviewer Discussion:**

During the rebuttal, reviewers raised concerns about the paper's novelty, efficiency, and experimental comparisons. The authors addressed some of these by emphasizing their contributions to improving diffusion-based decoders, providing additional quantitative and visual results, and clarifying the trade-off between inference time and quality, while proposing future optimizations. Although these responses clarified many points, some reviewers maintained concerns about the overall novelty and computational efficiency of the approach.

---

### Decision · Program_Chairs · 2025-01-22

Reject